# Digoxin Derivatives Sensitize a *Saccharomyces cerevisiae* Mutant Strain to Fluconazole by Inhibiting Pdr5p

**DOI:** 10.3390/jof8080769

**Published:** 2022-07-25

**Authors:** Daniel Clemente de Moraes, Ana Claudia Tessis, Rodrigo Rollin-Pinheiro, Jefferson Luiz Princival, José Augusto Ferreira Perez Villar, Leandro Augusto Barbosa, Eliana Barreto-Bergter, Antônio Ferreira-Pereira

**Affiliations:** 1Departamento de Microbiologia Geral, Instituto de Microbiologia Paulo de Góes, Centro de Ciências da Saúde, Universidade Federal do Rio de Janeiro, Cidade Universitária, Avenida Carlos Chagas Filho 373, Rio de Janeiro 21941-590, Brazil; danielcmoraes@micro.ufrj.br (D.C.d.M.); rodrigorollin@gmail.com (R.R.-P.); eliana.bergter@micro.ufrj.br (E.B.-B.); 2Instituto Federal do Rio de Janeiro, Rua Senador Furtado 121, Rio de Janeiro 20270-021, Brazil; atessis@gmail.com; 3Laboratory of Cell Biochemistry, Universidade Federal de São João del Rei, Campus Centro-Oeste Dona Lindu, Avenida Sebastião Gonçalves Coelho 400, Divinópolis 35501-296, Brazil; princivalj@ufsj.edu.br (J.L.P.); zevillar@ufsj.edu.br (J.A.F.P.V.); lbarbosa.ufsj@gmail.com (L.A.B.)

**Keywords:** digoxin, efflux pump, fluconazole, Pdr5p, *Saccharomyces cerevisiae*, yeast

## Abstract

The poor outcome of treatments for fungal infections is a consequence of the increasing incidence of resistance to antifungal agents, mainly due to the overexpression of efflux pumps. To surpass this mechanism of resistance, a substance able to inhibit these pumps could be administered in association with antifungals. *Saccharomyces cerevisiae* possesses an efflux pump (Pdr5p) homologue to those found in pathogenic yeast. Digoxin is a natural product that inhibits Na^+^, K^+^-ATPase. The aim of this study was to evaluate whether digoxin and its derivatives (i.e., DGB, digoxin benzylidene) can inhibit Pdr5p, reversing the resistance to fluconazole in yeasts. An *S. cerevisiae* mutant strain that overexpresses Pdr5p was used in the assays. The effects of the compounds on yeast growth, efflux activity, and Pdr5p ATPase activity were measured. All derivatives enhanced the antifungal activity of fluconazole against *S. cerevisiae*, in comparison to fluconazole alone, with FICI values ranging from 0.031 to 0.500. DGB 1 and DGB 3 presented combined effects with fluconazole against a *Candida albicans* strain, with fractional inhibitory concentration index (FICI) values of 0.625 and 0.281, respectively The compounds also inhibited the efflux of rhodamine 6G and Pdr5p ATPase activity, with IC_50_ values ranging from 0.41 μM to 3.72 μM. The results suggest that digoxin derivatives impair Pdr5p activity. Considering the homology between Pdr5p and efflux pumps from pathogenic fungi, these compounds are potential candidates to be used in association with fluconazole to treat resistant fungal infections.

## 1. Introduction

Treating patients affected by fungal infections has become arduous for clinicians due to the increasing incidence of resistance to antifungal drugs, especially those belonging to the azole class [1]. Moreover, the low number of drugs available for treatment, the toxicity related to their use, and the severity of the diseases—especially in immunocompromised individuals—jeopardize the prognosis of the treatment [2]. Thus, it is imperative that new pharmacological therapies be designed to allow the proper management of patients suffering from fungal infections. 

Fluconazole was the first triazole antifungal agent approved by the Food and Drug Administration (FDA), and has been considered the first-choice drug to treat fungal infections (except onychomycosis) since the 1990s, due to its high bioavailability and low toxicity [3]. Nonetheless, its overuse for almost 30 years has boosted the rise of multidrug-resistant (MDR) strains. These organisms show resistance not only to fluconazole, but also to several unrelated drugs, such as morpholines, cycloheximide, anisomycin, and doxorubicin, among others [4]. One of the main mechanisms of MDR is associated with the overexpression of efflux pumps within the fungal plasma membrane [5]. These proteins extrude the drug from the intracellular environment, using either ATP hydrolysis (ATP-binding cassette (ABC) pumps) [6] or an electrochemical gradient (major facilitator superfamily (MFS) pumps) [7] as energy source, thereby avoiding its accumulation inside the cytoplasm and, thus, precluding its pharmacological activity. Since MDR proteins are predominantly responsible for the failure of fluconazole-based treatments, inhibiting their activity would allow the antifungal agent to exert its optimal activity. 

*Saccharomyces cerevisiae* is a ubiquitous yeast, and is considered to be non-pathogenic. Nonetheless, some studies have reported bloodstream infections caused by this microorganism [8,9]. Moreover, *S. cerevisiae* possesses several MDR proteins, and Pdr5p is currently the best studied of them [10]. This transporter shares a high similarity with those found in pathogenic fungi, such as *Candida* spp. [11]. Since *S. cerevisiae*’s genetics are thoroughly understood, and its genes can be easily silenced or overexpressed, it is used as a tool to study efflux transporters belonging to the MDR family. Decottignies et al. (1998) constructed *S. cerevisiae* strains that overexpress specific MDR transporters, including one strain that overexpresses Pdr5p. In addition, one strain was constructed with the deletion of all genes related to the MDR phenotype. Using these strains, it was possible to evaluate the effects of compounds on a specific transporter. Another advantage is that the resistance does not depend on selective pressure [12].

Digoxin is a cardiotonic glycoside extracted from *Digitalis* sp. used to treat heart failure, atrial fibrillation, and other coronary conditions. Its mechanism of action is related to the inhibition of Na^+^, K^+^-ATPase—an ATP-driven protein that transports Na^+^ and K^+^ ions across the plasma membrane [13]. Considering that most MDR fungal proteins also use ATP hydrolysis as an energy source, digoxin and its derivatives could be able to inhibit their activity, reversing the MDR phenotype. There are no reports of the antifungal activity of these substances. Alves et al. (2015) observed that digoxin and its derivatives presented anticancer activity against human cervix carcinoma (HeLa cell line) and human colon carcinoma (RKO-AS45-1 cell line). Moreover, the compounds inhibited Na/K-ATPase from human kidney and rat brain preparations [14]. 

Due to the severity of fungal infections, and the complexity of overcoming those related to resistant microorganisms, this study aimed to evaluate the effects of digoxin and its synthetic derivatives on the functioning of fungal MDR transporters, using a resistant *Saccharomyces cerevisiae* strain overexpressing the efflux pump Pdr5p as a model for study.

## 2. Materials and Methods

### 2.1. Strains and Culture Conditions 

In this study, two *S. cerevisiae* mutant strains were used. Strain AD/124567 (genotype: *MATα*, *PDR1-3*, *ura3*, *his1*, *Δyor1::hisG*, *Δsnq2::hisG*, *Δpdr10::hisG*, *Δpdr11::hisG*, *Δycf1::hisG*, *Δpdr3::hisG*) overexpressed Pdr5p, but the genes of all other ABC transporters related to the MDR phenotype (Yor1p, Snq2p, Pdr10p, Pdr11p, and Ycf1p) were deleted. A second strain, AD/1234567, had all MDR related genes deleted, including *PDR5* (genotype: *MATα*, *PDR1-3*, *ura3*, *his1*, *Δyor1::hisG*, *Δsnq2::hisG*, *Δpdr5::hisG*, *Δpdr10::hisG*, *Δpdr11::hisG*, *Δycf1::hisG*, *Δpdr3::hisG*) [12]. Therefore, this strain was highly sensitive to fluconazole. One fluconazole-resistant *Candida albicans* strain was also used in this study, namely, 95-142. This strain was isolated from the throat of a patient and identified by CHROMagar^TM^ and PCR. The isolate overexpressed CaCdr1p and CaCdr2p (the most important MDR transporters of *C. albicans*), and was kindly provided by Dr. Theodore White (University of Missouri, Kansas City, MO, USA) [15]. 

Yeasts were cultured in yeast peptone dextrose (YPD) medium (1% yeast extract, 2% peptone, 2% glucose) at 30 °C (*S. cerevisiae*) or 37 °C (*C. albicans*) with agitation (100 rpm), and harvested in the exponential growth phase (optical density at 600 nm = 1.0). 

### 2.2. Chemicals

Fluconazole (generic, reagent grade; purity ≥ 99%) was purchased from the university pharmacy of the Universidade Federal de Juiz de Fora (UFJF, Juiz de Fora, MG, Brazil). Stock solutions at 2 mg/mL were prepared in distilled water, sterilized by filtration (0.22 μm), and stored at –20 °C. The tested digoxin compounds—DBG1, DGB2, DGB3, DGB4, DGB5, DGB6, and DGB7 (Figure 1)—were synthesized as described by Alves et al. (2015) [14] and solubilized in dimethyl sulfoxide (DMSO) (Sigma Aldrich^®^, St. Louis, MO, USA) at a final concentration of 10 mM. Molecular weights and calculated values of LogP (ChemDraw Ultra v 12.0^®^, Cambridge Soft) are shown in Table 2. Digoxin derivatives present molecular weights ranging from 869 g/mol to 938 g/mol, while cLogP ranges from 2.96 to 5.09. Rhodamine 6G (R6G), RPMI 1640, sorbitol, sodium azide, and 2-mercaptoethanol were also purchased from Sigma-Aldrich^®^. KH_2_PO_4_, NaCl, Na_2_HPO_4_, and glucose were purchased from VETEC (Duque de Caxias, RJ, Brazil). Yeast extract was purchased from KASVI (São José dos Pinhais, PR, Brazil). Peptone was purchased from HiMedia (Mumbai, Maharashtra, India). FK-506 was purchased from Tecoland (Irvine, CA, USA). Zymolyase was purchased from Thermo Fisher Scientific (Waltham, MA, USA).

### 2.3. Antifungal Susceptibility Test

The antifungal activity of the compounds was evaluated via a microbroth dilution method described by Niimi et al. [16]. Cell suspensions from AD/1234567 and AD/124567 were added to YPD medium (final concentration = 2 × 10^4^ cells/mL) and incubated at 30 °C in the presence of 2-fold serial dilutions of digoxin and its derivatives (100–0.39 µM) for 48 h with agitation (final volume = 200 μL; 75 rpm). Culture growth was measured using a microplate reader at 600 nm (Fluostar Optima, BMG Labtech, Offenburg, Germany). Cells were also incubated in the absence of the compounds (growth control). Growth inhibition was calculated as follows:
$$\%\;\text{Growth}\;\text{inhibition}\;=\;\frac{(\text{Absorbance}\;\text{of}\;\text{growth}\;\text{control}\;-\;\text{Absorbance}\;\text{of}\;\text{treated}\;\text{sample})\;\times \;100}{\text{Absorbance}\;\text{of}\;\text{growth}\;\text{control}}$$

### 2.4. Disk Diffusion Assay

To evaluate the ability of the compounds in reversing the MDR phenotype of strain AD/124567, a chemosensitization assay was performed, with slight modifications [16]. Briefly, YPD medium containing 150 µg/mL fluconazole was prepared with 1.5% agarose, and a cell suspension from AD/124567 was incorporated into molten medium (45 °C), reaching a final concentration of 2.4 × 10^6^ yeasts/mL. Subsequently, 50 μg of each compound was applied to 6-milimeter Whatman 3MM^®^ blotting paper disks (Sigma-Aldrich^®^, St. Louis, MO, USA), dried at 30 °C for 30 min, and placed on the solidified medium surface. The plates were incubated at 30 °C for 48 h, photographed, and the inhibition zones were measured. A fluconazole-free medium was also used to evaluate the effects of the compounds alone on yeast growth.

### 2.5. Checkerboard Assay 

The effects of digoxin derivatives combined with fluconazole were also evaluated in liquid medium using a checkerboard assay [16], with slight modifications (instead of using liquid CSM-URA, yeasts were incubated in YPD or RPMI 1640). Briefly, cell suspensions of the AD/124567 strain (final concentration = 2 × 10^4^ yeasts/mL) and the 95-142 strain (final concentration = 5 × 10^3^ yeasts/mL) were inoculated into YPD medium at 30 °C or RPMI 1640 medium at 37 °C for 48 h, with agitation (75 rpm), in the presence or absence of different concentrations of digoxin derivatives (0.195–100 μM) and fluconazole (7.5–480 μg/mL). Culture growth was measured using a microplate reader at 600 nm (Fluostar Optima, BMG Labtech, Offenburg, Germany). Growth inhibition was calculated as described in Section 2.3. The concentrations of digoxin derivatives alone and fluconazole alone able to inhibit 80% of yeast growth were defined as the MIC (minimum inhibitory concentration) of each drug. The concentrations of the combinations of digoxin derivatives with fluconazole that inhibited 80% of yeast growth were defined as the MIC of each pair of combined drugs. The interpretation of results obtained from the combinations was based on the fractional inhibitory concentration index (FICI), which is determined by the following equation:FICI = digoxin derivative FIC + fluconazole FIC(1)

A digoxin derivative’s FIC (fractional inhibitory concentration) is defined as its MIC combined/MIC alone, while fluconazole’s FIC is defined as fluconazole’s MIC combined/fluconazole’s MIC alone. FICI ≤ 0.5 points to a synergic interaction, whereas 0.5 < FICI < 4.0 and FICI > 4 indicate indifferent and antagonistic interactions, respectively.

### 2.6. Rhodamine 6G Accumulation Assay

Efflux assays were performed as described by Reis de Sá et al. (2017) [17]. Briefly, AD/124567 cells in the exponential growth phase were harvested by centrifugation at 5000 rpm at room temperature for 5 min, and then washed twice with phosphate-buffered saline (PBS) (0.43 g of KH_2_PO_4_; 7.2 g of NaCl; 1.85 g of Na_2_HPO_4_; pH 7.2; 990 mL of distilled water). Cells were then resuspended in 10 mL of PBS and starved for 2 h at 4 °C. Subsequently, 10^7^ cells/mL were incubated for 30 min at 30 °C with 15 μM R6G. After incubation, cells were washed twice with PBS and further incubated in the presence of the compounds, DMSO, and FK-506 (a classical inhibitor of ABC transporters) for 60 min at 30 °C. The cells were then incubated with glucose at a final concentration of 0.2% for 30 min at 30 °C, and then pelleted by centrifugation at 9000 rpm for 2 min at room temperature. The supernatants were transferred to the wells of a 96-well polystyrene plate, and fluorescence was measured with a plate reader (Fluostar Optima, BMG Labtech, Offenburg, Germany) at 485 nm (excitation) and 538 nm (emission). To evaluate R6G efflux, fluorescence emitted by the dye in the extracellular milieu was measured. Thus, the fluorescence was directly related to extracellular R6G concentration, and then to Pdr5p activity. AD/1234567 was used as a control. R6G efflux was calculated as follows:
(2)$$\%\;\text{efflux}\;=\;\frac{\text{Fluorescence}\;\text{of}\;\text{treated}\;\text{sample}\;\times \;100}{\text{Fluorescence}\;\text{of}\;\text{AD}/124567\;\text{with}\;\text{glucose}}$$

After removal of the supernatant for fluorescence measurement, pellets were resuspended in PBS and visualized by fluorescence microscopy (Nikon Eclipse E400, Nikon, Tokyo, Japan).

### 2.7. Preparation of Plasma Membranes

Pdr5p is an ABC transporter, using ATP hydrolysis as energy source to extrude drugs from the cell. Plasma membranes enriched with Pdr5p were obtained to assess whether the compounds inhibit the ATPase activity of the transporter. Cells from AD/1234567 and AD/124567 strains in the exponential growth phase were washed with 10 mM sodium azide. Yeast cell walls were then digested by incubation at 37 °C for 60 min with 100 KU/g zymolyase (4.1 mg to each 10^7^ cells) and 2-mercaptoethanol (58 μL in 15 mL of zymolyase buffer: 2.8 M sorbitol, 0.1 M KH_2_PO_4_, 10 mM sodium azide). Unlysed cells and debris were removed by centrifugation at 4500× *g* for 10 min at 4 °C. The cleared supernatant was then centrifuged at 12,000× *g* for 40 min at 4 °C to remove organelles, such as mitochondria, and then at 20,000× *g* for 40 min at 4 °C to generate the plasma membrane fractions. The plasma membranes were then stored in liquid nitrogen [18].

### 2.8. ATPase Activity

The effects of the compounds on the ATPase activity of Pdr5p were assessed by incubating purified membranes (0.013 mg/mL) obtained from the AD/1234567 and AD/124567 strains for 1 h at 37 °C in a reaction medium (100 mM Tris-HCl pH 7.5, 4 mM MgCl_2_, 75 mM KNO_3_, 7.5 mM NaN_3_, 0.3 mM (NH_4_)_6_Mo_7_O_24_, and 3 mM ATP) in the presence of serial dilutions of the compounds (100–0.39 μM). The reaction was stopped by adding 1% sodium dodecyl sulfate (SDS). The inorganic phosphate generated was measured by the Fiske–Subbarow method. Purified plasma membranes from AD/1234567 were used as negative controls [18]. The concentration of the compounds able to inhibit 50% of ATPase activity was defined as the IC_50_.

### 2.9. Hemolysis Assay

The effects of the compounds on the integrity of red blood cells obtained from sheep were evaluated as described by Niimi et al. (2004) [16]. Cells were washed three times and resuspended in PBS to a final of concentration of 2% v/v. The cells were then incubated in the presence of different concentrations of DGB 1-7 (128–0.5 μM) for 60 min at 37 °C. Afterwards, cells were harvested by centrifugation at 3000× *g* for 5 min at room temperature, and the absorbance of the hemoglobin released in the supernatant due to hemolysis was measured at 540 nm (Fluostar Optima, BMG Labtech, Offenburg, Germany). Thus, absorbance at 540 nm is directly related to hemolysis. Controls of 100% and 0% hemolysis were performed by incubating the cells in PBS in the presence or absence of 1% Triton X-100, respectively. A control with DMSO was also performed. Hemolysis was calculated as follows:
(3)$$\%\;\text{Hemolysis}\;=\;\frac{\text{Absorbance}\;\text{of}\;\text{treated}\;\text{sample}\;\times \;100}{\text{Absorbance}\;\text{of}\;\text{Triton}\;\text{X-100}\;\text{control}}$$

The use of ovine blood cells in this study was approved by the Ethics Committee for the Use of Animals in Research of the Universidade Federal do Rio de Janeiro (CEUA-UFRJ: 157/21)

### 2.10. Statistical Analysis

All experiments were performed three times. Data were analyzed by Student’s *t*-test, and *p*-values lower than 0.05 were considered significant. 

## 3. Results

### 3.1. Antifungal Susceptibility Test

The antifungal activity of the compounds alone was evaluated to determine their toxic concentrations against *S. cerevisiae*. Only DGB3 presented antifungal activity against the tested strains. At 100 μM, this derivative inhibited the growth of AD/1234567 and AD/124567 by 50% and 80%, respectively. 

### 3.2. Disk Diffusion Assay

To verify whether the substances enhanced the antifungal activity of fluconazole, a disk diffusion assay was performed. None of the substances inhibited the growth of AD/124567 in the absence of fluconazole (Figure 2A). On the other hand, in the presence of fluconazole, inhibition zones (DGB1: 8.8 mm, DGB2: 14.0 mm, DGB3: 12.4 mm, DGB4: 11.2 mm, DGB5: 19.2 mm, DGB6: 10.0 mm, DGB7: 14.8 mm, FK-506: 18 mm) were observed around the disks containing each substance, except for digoxin and DMSO (Figure 2B).

### 3.3. Checkerboard Assay

A checkerboard assay was performed to evaluate the combinatorial activity between the tested compounds and fluconazole. Against the strain AD/124567, all digoxin derivatives interacted synergistically with fluconazole, with FICI values varying from 0.031 to 0.500. The digoxin derivatives decreased fluconazole’s MIC by 4–64-fold. When tested against the 95–142 strain, only DGB1 and DGB3 interacted with fluconazole, presenting FICI values of 0.625 and 0.281, respectively (Table 1). DGB1 and DGB3 led to a twofold and fourfold decrease in fluconazole’s MIC for the strain 95–142, respectively. Except for DGB3 against AD/124567, none of the compounds inhibited the growth of the strains. Thus, the MIC values were considered as > 100 μM. For the calculation of FICI and the determination of the nature of interactions, an MIC of 200 μM was considered for all compounds. Our data show that the compounds indeed enhance the antifungal activity of fluconazole against highly resistant fungal strains.

### 3.4. Rhodamine 6G Accumulation Assay

To investigate whether the combinatory effects of the compounds occurred due to Pdr5p inhibition, assays with rhodamine 6G were performed. In the absence of glucose, the strain AD/124567 (which overexpresses Pdr5p) accumulated R6G in the cytoplasm (Figure 3A), as did the strain AD/1234567 (which does not overexpress any efflux transporters) (Figure 3B). The addition of glucose allowed the efflux of R6G by Pdr5p (Figure 3C). Glucose was used because it is metabolized by yeasts generating ATP, thus activating Pdr5p via ATP hydrolysis. All digoxin derivatives inhibited R6G efflux in the presence of glucose (Figure 3D–J). On the other hand, digoxin was unable to inhibit Pdr5p activity (Figure 3K). Table 2 and Figure 4 describe the rates of R6G efflux after treatment with the compounds. Except for DGB4 and digoxin, all substances inhibited R6G efflux by more than 90%. These data reinforce the fact that the synergism between digoxin derivatives and fluconazole is a consequence of Pdr5p inhibition.

### 3.5. ATPase Activity

The ATPase activity of Pdr5p was measured to verify whether the compounds could inhibit this transporter by blocking its enzymatic activity. All of the compounds inhibited the ATPase activity in a dose-dependent manner (Figure 5), with IC_50_ values ranging from 0.41 μM to 3.72 μM (DGB1: 1.25 μM, DGB2: 0.49 μM, DGB3: 2.13 μM, DGB4: 2.19 μM, DGB5: 0.41 μM, DGB6: 0.53 μM, DGB7: 3.72 μM) (Table 2). The results show that digoxin derivatives prevent Pdr5p activity by removing its energy source.

### 3.6. Hemolysis Assay

A hemolysis assay was performed to conduct a preliminary evaluation of compound toxicity. Even at high concentrations (128 μM), the hemolytic activity of the compounds was comparable to that of the control with PBS (Figure 6). Drugs for the treatment of systemic infections need to reach the bloodstream. Thus, it is essential that they do not exert hemolytic effects. The data reported here show that digoxin derivatives fulfill this requirement, and may therefore be good candidates in the search for new agents against resistant infections.

## 4. Discussion

Overcoming antimicrobial resistance is one of the greatest challenges when dealing with infectious diseases [19]. Currently, only three classes of antifungals are available to treat systemic infections, and there is a worrisome increase in resistance to these drugs. Moreover, during the COVID-19 pandemic, an increase in deaths caused by fungi—mainly *Aspergillus* spp. and *Candida* spp.—was observed. [20]. In this context, the development of new strategies to manage fungal infections is crucial—especially for infections caused by fungi resistant to classic antifungal drugs.

*Saccharomyces cerevisiae* is a yeast that is widely employed as a model organism in biochemistry and microbiology, since it is non-pathogenic and easily manipulated at the genetic level [12]. Considering the study of antimicrobial resistance, using *S. cerevisiae* as a tool is interesting because this yeast expresses transporters related to the MDR phenotype, such as Pdr5p, Pdr10p, Pdr11p, Pdr12p, Pdr15p, Pdr18p, Ycf1p, Snq2p, and Yor1p [8]. Pdr5p is the most studied of these transporters, due to its high promiscuity [21] and homology with efflux transporters found in pathogenic fungi, such as *Candida* spp. [11] and *Aspergillus* spp. [22]. 

Considering the relevance of efflux transporters in antifungal resistance, and the suitability of using *S. cerevisiae* as a study model, we evaluated the ability of digoxin and its derivatives to inhibit Pdr5p and, consequently, reverse the MDR phenotype.

Firstly, the antifungal activity of the compounds was evaluated, and the data showed that DGB3 presented a weak activity. Nonetheless, to the best of our knowledge, this is the first study to report the antifungal activity of digoxin or analogous molecules. The compounds tested presented antifungal activity at higher concentrations; however, it should be taken into consideration that digoxin has a narrow therapeutic window [23]. Since there is so far no pharmacokinetic characterization of its derivatives, it may be advisable to assume that their toxic profiles are similar to that of digoxin. Thus, even if the compounds possess antifungal activity at concentrations higher than 100 μM, it would probably not be possible to use them in clinical settings.

The combinatory activity of the compounds with fluconazole against *S. cerevisiae* growth was evaluated by agar diffusion assay and checkerboard titration. On solid media, none of the substances presented antifungal activity on their own. However, co-treatment with fluconazole inhibited the growth of *S. cerevisiae* in both assays. Only digoxin was indifferent to interaction with fluconazole. These results indicate that inserting a benzylidene group in the lactone ring of digoxin is essential to allow the substances to enhance the antifungal activity of fluconazole. Frohock et al. (2020) observed that 5-benzylidene-4-oxazolidinones improved the activity of seven antibacterial agents against *Staphylococcus aureus* [24]. Moreover, the most active compound—namely, 24—presented a longer alkyl side chain. In our study, we observed that the compound with the highest FICI value (DGB1) was also the one with the lowest molecular weight. Nonetheless, there was no relationship between the molecular weight of the derivatives and their synergism with fluconazole, since substances with similar molecular weights—such as DGB1, DGB2, and DGB4—presented different FICI values. Moreover, hydrophobicity does not explain the differences in the activities of the compounds. The derivatives DGB1, DGB2, DGB4, and DGB5 presented cLogP between 3.78 and 3.98, but showed different FICI values, ranging from 0.032 to 0.500. 

Digoxin is a substrate of P-glycoprotein (Pgp)—a human transporter protein associated with the MDR phenotype [25]. Pdr5p and Pgp share structural similarities [26], and it was hypothesized that digoxin could also be a substrate for Pdr5p, therefore competitively inhibiting this transporter. Except for digoxin, all compounds affected rhodamine 6G efflux which, in turn, was consistent with the results obtained in the previous experiments. Thus, digoxin may not be transported by Pdr5p. It cannot be excluded, however, that digoxin derivatives inhibit R6G efflux by binding to the same catalytic site as R6G. Previously, our group described the inhibitory effect of natural and synthetic compounds on Pdr5p. Oroidin, a sponge-derived alkaloid, inhibited R6G efflux by 60% at 200 μM [27]. Beta-lapachone, a naphthoquinone obtained from *Tabebuia* sp., inhibited Pdr5p activity by 79.4% at 100 μg/mL [28]. In this study, it was observed that digoxin derivatives inhibited R6G efflux by 76.65–100%. Moreover, digoxin did not affect Pdr5p activity. Drug repurposing—i.e., discovering new uses for drugs that have already been approved to treat other diseases—has been a widely explored tool to develop antifungal treatments, since it shortens the time needed for drug approval [29]. Data show that digoxin neither presented antifungal activity nor inhibited Pdr5p. However, this study shows that known drugs can be used as scaffolds to develop novel molecules with biological activities. In addition to competitive inhibition, compounds could block Pdr5p activity by impairing its ATPase activity. Indeed, digoxin derivatives inhibited ATP hydrolysis, with very low IC_50_ values. This assay utilizes purified plasma membrane; thus, it may be assumed that the tested substances directly affect ATPase activity. Oliveira et al. (2021) observed that digoxin derivatives at 8–35 μM inhibited the Na/K-ATPase activity of cancer cells by 28–56% [30]. Other digoxin derivatives inhibited Na/K-ATPase from human kidney cells at 0.34–11 μM [14]. The data obtained in this study show that digoxin derivatives inhibit Pdr5p ATPase activity with IC_50_ values comparable to those observed for Na/K-ATPase. These results corroborate the efflux inhibition observed in the previous experiment. Interestingly, digoxin inhibits Na/K-ATPase with lower IC_50_ values than its derivatives [14], but exerts no effect on Pdr5p. In silico studies should be conducted to unveil why a benzylidene group is needed to inhibit Pdr5p ATPase activity. Furthermore, this would allow the synthesis of novel digoxin derivatives with more potent activities.

In addition, a compound may disturb ATP hydrolysis by decreasing mitochondrial membrane potential (MMP). In this study, depolarization of MMP was not observed after treatment of *S. cerevisiae* with digoxin derivatives. Again, neither molecular weight nor hydrophobicity explained the differences observed between the compounds. Interestingly, the compound with the highest IC_50_ was DGB7—the most active derivative in the checkerboard assay. Although this study is focused on evaluating efflux pump inhibition, other mechanisms could be involved in the synergism between digoxin derivatives and fluconazole. 

As mentioned above, the strain AD/124567 was used in this study because it overexpresses Pdr5p—a transporter homologue to those found in pathogenic fungi [11,22]. To assess whether the ability of a compound to enhance fluconazole activity could be extrapolated to other microorganisms, a checkerboard assay using *Candida albicans* strain 95–142 was performed. In contrast to the observations with *S. cerevisiae*, only two substances improved fluconazole’s activity against 95–142 growth. One hypothesis is that the composition of the plasma membrane or cell wall of *C. albicans* hampered the entry of the compounds into the cytoplasm [31]. A second hypothesis is that Pdr5p is more similar to CaCdr1p than to CaCdr2p [32], and that 95–142 overexpresses both transporters. Thus, compounds could be able to inhibit CaCdr1p, but CaCdr2p functioning may be sufficient for *C. albicans* to extrude fluconazole from the intracellular milieu.

Finally, a hemolysis assay was performed to verify the toxicity of digoxin derivatives against erythrocytes. None of the compounds were toxic to these cells, including digoxin. However, it is known that cardiotonic glycosides are toxic due to the inhibition of Na/K-ATPase [33]. Although no hemolytic effects have been observed, it is necessary to evaluate the toxicity of digoxin derivatives in vivo, using more complex organisms. 

## 5. Conclusions

This study shows for the first time that digoxin derivatives inhibit the activity of Pdr5p—a *S. cerevisiae* multidrug transporter—sensitizing this fungus to fluconazole. Moreover, two derivatives (i.e., DGB1 and DGB3) improved the antifungal activity of fluconazole against a *C. albicans* strain that was resistant due to an efflux mechanism. These results are promising, considering that the compounds were active at low concentrations. Considering the homology between Pdr5p and efflux pumps from pathogenic fungi, along with the increasing incidence of *S. cerevisiae* infections, digoxin derivatives appear as potential candidates to be used in association with fluconazole to treat resistant fungal infections, Further studies should be conducted to assess the in vivo toxicity of these substances. Moreover, the activity of DGB1 and DGB3 combined with fluconazole should be evaluated against a larger set of *Candida* spp. strains.

## Figures and Tables

**Figure 1 jof-08-00769-f001:**
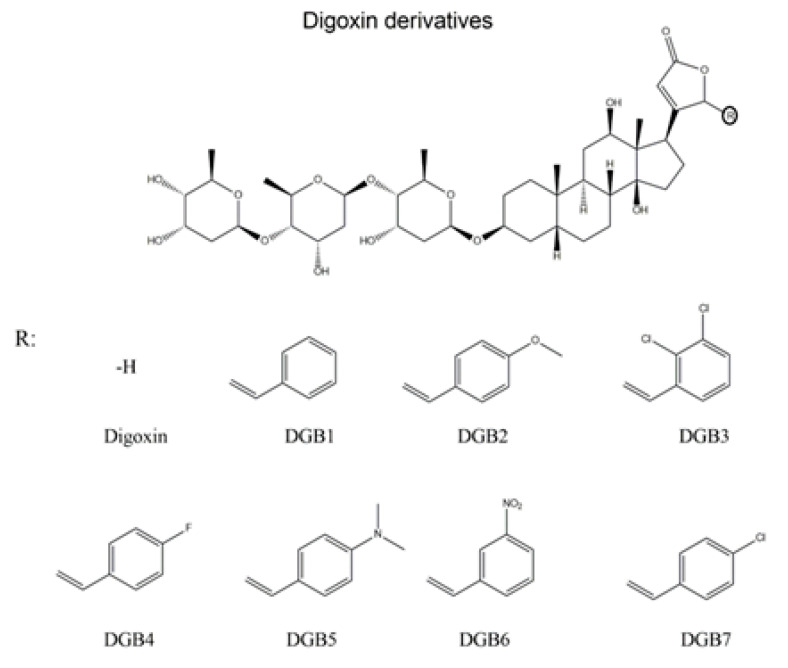
Structure of digoxin and its derivatives.

**Figure 2 jof-08-00769-f002:**
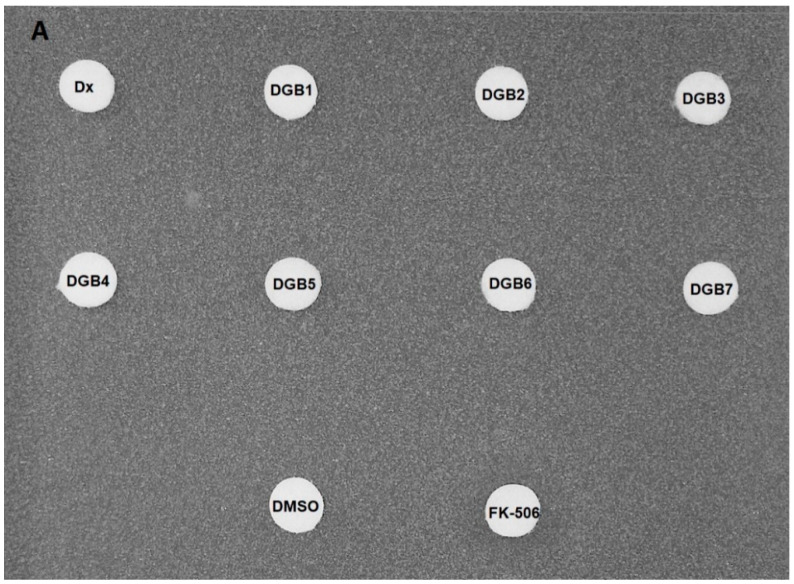
Chemosensitization assay by disk diffusion method: Digoxin (Dx) and its derivatives (DGB1–DGB7) were added to filter paper disks and incubated on plates containing the strain AD/124567 in the (**A**) absence or (**B**) presence of fluconazole (150 μg/mL). Dimethyl sulfoxide (DMSO) and FK-506 were used as controls. These images are representative of three independent experiments.

**Figure 3 jof-08-00769-f003:**
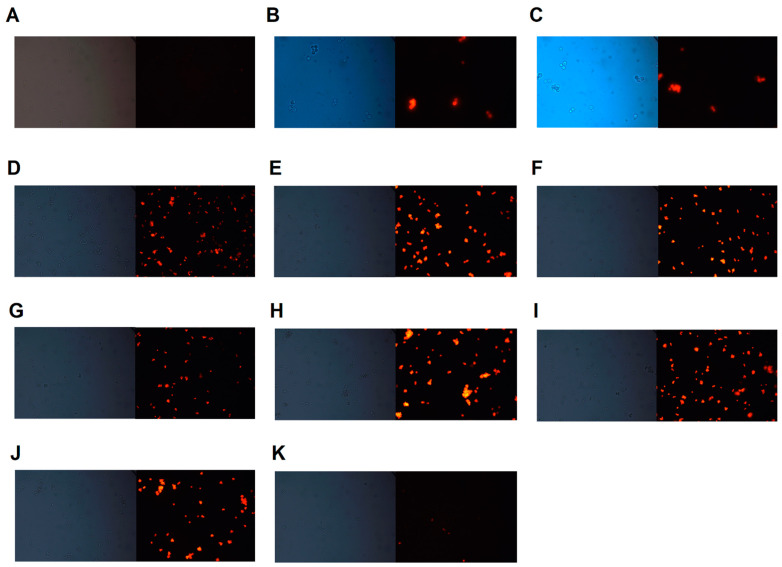
Rhodamine 6G accumulation assay: Cells were incubated in the presence or absence of digoxin and its derivatives, loaded with rhodamine 6G, and incubated with 0.2% glucose. The cells were then visualized by fluorescence microscopy. (**A**) Strain AD/1234567 without treatment and with glucose; (**B**) strain AD/124567 without treatment and without glucose; (**C**) strain AD/124567 + glucose; (**D**) strain AD/124567 + DGB1 + glucose; (**E**) strain AD/124567 + DGB2 + glucose; (**F**) strain AD/124567 + DGB3 + glucose; (**G**) strain AD/124567 + DGB4 + glucose; (**H**) strain AD/124567 + DGB5 + glucose; (**I**) strain AD/124567 + DGB6 + glucose; (**J**) strain AD/124567 + DGB7+ glucose; (**K**) strain AD/124567 + digoxin + glucose. Magnification: 400×. The images are representative of three independent experiments. All pictures are listed in normal size at Appendix A.

**Figure 4 jof-08-00769-f004:**
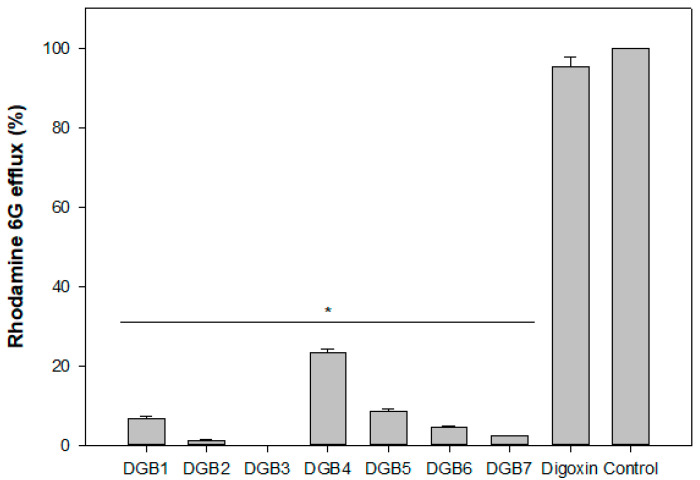
Determination of Rhodamine 6G efflux: Cells (strain AD/124567) were incubated in the presence or absence of digoxin and its derivatives, loaded with rhodamine 6G, and incubated with 0.2% glucose. The cells were then centrifuged, and the fluorescence of rhodamine 6G in the supernatant was measured. Data refer to three independent experiments. Results are expressed as % of efflux in comparison to controls (untreated cells); (*) *p* < 0.05 in comparison to controls. DGB 1-7: digoxin derivatives.

**Figure 5 jof-08-00769-f005:**
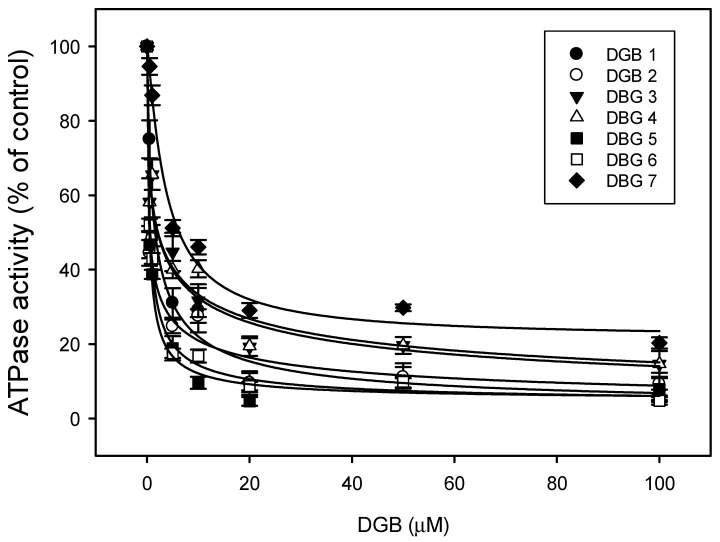
Effects of compounds on ATPase activity: Purified membranes enriched with Pdr5p were incubated in the presence of serial concentrations of digoxin derivatives, and the inorganic phosphate released was measured by the Fiske–Subbarow method. Data refer to three independent experiments. DGB 1-7: digoxin derivatives.

**Figure 6 jof-08-00769-f006:**
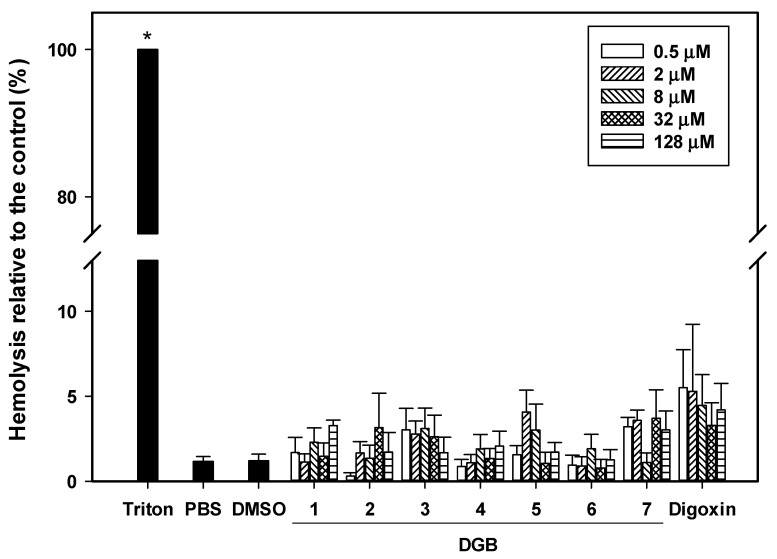
Hemolysis assay: Erythrocytes were incubated in the presence of serial concentrations of digoxin and its derivatives, and the absorbance of released hemoglobin was measured at 540 nm. Triton X-100 and PBS were used as controls of 100% and 0% hemolysis, respectively. DMSO was also used as a control. Data refer to three independent experiments; (*) *p* < 0.05 in comparison to PBS treatment. DGB 1-7: digoxin derivatives. PBS: phosphate-buffered saline. DMSO: dimethyl sulfoxide.

**Table 1 jof-08-00769-t001:** Checkerboard assay of digoxin derivatives against strains AD/124567 and 95–142.

Strain and Compound	Compound (µM)			Fluconazole (µg/mL)				
	MIC a	MIC c	FIC	MIC a	MIC c	FIC	FICI	Outcome
AD/124567								
DGB1	200	50	0.250	480	120	0.250	0.500	S
DGB2 DGB3 DGB4 DGB5 DGB6 DGB7	200 100 200 200 200 200	25 6.25 50 3.125 50 3.125	0.125 0.063 0.250 0.016 0.250 0.016	480 480 480 480 480 480	7.5 7.5 15 7.5 60 7.5	0.016 0.016 0.031 0.016 0.125 0.016	0.141 0.078 0.281 0.031 0.375 0.031	S S S S S S
95–142								
DGB1	200	25	0.125	62.5	31.25	0.500	0.625	I
DGB2 DGB3 DGB4 DGB5 DGB6 DGB7	200 200 200 200 200 200	200 6.25 200 200 200 200	1 0.031 1 1 1 1	62.5 62.5 62.5 62.5 62.5 62.5	62.5 15.63 62.5 62.5 62.5 62.5	1 0.250 1 1 1 1	2 0.281 2 2 2 2	I S I I I I

MIC a: MIC of the substance alone; MIC c: MIC of the compound combined with a second drug; FIC: fractional inhibitory concentration (MIC c/MIC a); FICI: fractional inhibitory concentration index (sum of each drug’s FIC); S: synergism; I: indifference. DGB 1-7: digoxin derivatives.

**Table 2 jof-08-00769-t002:** Results of rhodamine 6G accumulation and ATPase activity assays, and physicochemical properties of digoxin derivatives. The assays were performed three times on different days.

Compound	Rhodamine 6G Efflux (%)	IC_50_ (μM)	Molecular Weight	cLogP
DGB1	6.84	1.25	869	3.88
DGB2	1.31	0.49	899	3.81
DGB3	0.00	2.13	938	5.09
DGB4	23.35	2.19	887	3.98
DGB5	8.52	0.41	912	3.78
DGB6	4.66	0.53	914	2.96
DGB7	2.45	3.72	903	4.49

## Data Availability

Not applicable.

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
