# Peer review of "Digoxin Derivatives Sensitize a Saccharomyces cerevisiae Mutant Strain to Fluconazole by Inhibiting Pdr5p"

_jof, 2022, doi:10.3390/jof8080769_

Round 1

Author Response

Answers and comments for Reviewer 1:

1)Abstract

The Abstract did not provide an informative and quantitative summary of the research undertaken (Please rewrite the Abstract).

Thank you for your comment. Abstract was rewritten as suggested.

Line 22: “Enhanced the antifungal activity”. What's the ratio of the increase? It was compared to what? This is not clear.

Thank you for your question. To better understanding this information was added to the abstract.

Line 24: Conclusion is missing in the abstract. What are the future perspectives like, for instance, the applicability in clinical practice?

Thank you for your comment and suggestion. A conclusion paragraph was added as suggested.

2) Introduction

Line 38: “several unrelated drugs” What drugs? You cite only one reference to say several drugs.

Thank you for your comment and question about this point. This information was added as suggested by the reviewer.

Line 40: It is not clear how reference 5 (Sauna et al 2008) supports the conclusion that "The main MDR mechanism is associated to the overexpression of efflux pumps within the plasma membrane of fungi."

Thank you for your comment and criticism about it. Reference 5 was replaced with a more suitable one.

Line 47 -50: The authors do not discuss conveniently the cell model they used, for instance they could discuss about the overexpresses efflux pump Pdr5p.

Thank you for your comment about cell model. More information was added as suggested by the reviewer.

Line 51: The authors should include some results from previous studies on this compound.

Thank you for your comment and suggestion. This information was added as suggested by the reviewer, at Introduction section.

3) Materials and Methods

The authors write superficial information on the different assays performed; it is necessary to describe the techniques used.

The following should be given due attention:

Lines 63 and 66: Please provide the complete genotype of the S. cerevisiae strains

Thank you for your comment. All information about genetic of strains was included as suggested by the reviewer.

Line 72: replace "yeasts were grown" by "yeasts were cultured"

Thank you for your suggestion and it was replaced as suggested by you.

Line 72-73: “yeast extract, peptone, glucose” please include the manufacturer of chemicals.

Thank you for your comment. This information was included as suggested by the reviewer.

Line 75: In Chemicals item, the authors should describe the origin of all chemical reagents used in assays, how stock solutions and reagents were made can be insert in the description of each method. It is not clear if fluconazole is the drug or a prescription product. Please review.

Thank you for your comment. The missing information was added to this section.

Line 83: Why did the authors use RPMI-1640 medium?

Thank you for your question. We used RPMI-1640 to perform checkerboard with C. albicans, because this medium is less nutritive than YPD. Since YPD presents a high amount of carbon and nitrogen sources, when C. albicans is incubated in this medium, even sensitive strains grow in the presence of fluconazole. Thus, using RPMI-1640 provides more reliable results. On the other hand, S. cerevisiae does not grow well in RPMI-1640, and then we use YPD with this yeast.

Line 88: Your method lacks information. Were controls made? Was cell growth directly related to absorbance?

Thank you for all your questions about it. This information was added to the manuscript.

 Lines 92-93: Replace "cell growth" by "culture growth". Check throughout the manuscript.

Thank you for your comment. This modification was made as suggested by the reviewer.

 Line 93: Since the authors used a microplate reader it is assumed that these assays were done in microplates. What was the final volume of cultures? Was 75 rpm enough for keeping cells in suspension?

Thank you for your comment. The final volume was 200 μl per well. Yes, cells were kept in suspension at 75 rpm.

Line 98: How ha  cell suspension? ha tis the final concentration? It is the final cell density that matters.

Thank you for your comment. This information was added to the manuscript.

Line 108: Why the concentrations of the yeasts are different?

Thank you for your question. In growth assays, we use cellular concentrations that allow us to obtain absorbance values of about 1.0 at positive control (without treatment). We use different cellular concentrations because, in our experience, Candida albicans (95-142 strain) multiplies faster than AD/124567 strain.

Line 110: What concentration of digoxin derivatives and fluconazole? This information is important.

Thank you for your question. This information was included as suggested by the reviewer.

Line 115: The authors need to provide the meaning of FIC and MIC, how did the authors calculate MIC values? This part is very confusing, please explain what is combined MIC?

Thank you for your comment. This information was added to the manuscript.

Line 122: At what growth phase of the culture cells were harvested?

Thank you for your comment. This information was included as suggested by the reviewer.

Line 123: “KH2PO4; NaCl; Na2HPO” please address the company of the origin for any chemicals.

Thank you for your comment. This information was included as suggested by the reviewer.

Line 127: FK-506 is not listed in the chemicals section. You should provide that information.

Thank you for your comment. This information was included as suggested by the reviewer.

Line 129: What is the temperature for centrifugation?

Thank you for your comment. This information was included as suggested by the reviewer.

Line 135: Did you centrifuge the suspension? It is not mentioned.

Thank you for your comment. The suspension was centrifuged, the pellet was used in microscopy and supernatant was used to measure rhodamine 6G efflux.

 Line 141: “exponential growth phase” specify, what is the cell density or absorbance?

Thank you for your comment. This information was included as suggested by the reviewer at section 2.1.

Line 142: “sodium azide, zymolase, 2-mercaptoethanol” please address the company of the origin for any chemicals. Check all manuscript for the suppliers of chemicals.

Thank you for your comment. This information was included as suggested by the reviewer at section 2.2.

Line 142: The information on the units you used to make spheroplasts is missing. The buffer for the zymolyase is also missing. This is important in order to know whether or not cell lysis was efficient.

Thank you for your comment. This information was included in the manuscript.

 Line 143: ha tis the condition for centrifugation?

Thank you for your comment. This information was included in the manuscript.

Line 155: The method should not be named “viability” since the method only measures hemolysis, not viability.

Thank you for your comment. The term “viability” was changed to “integrity”.

Line 160: ha tis the temperature for centrifugation?

Thank you for your comment. This information was included in the manuscript.

 Line 160: Use italic in gravitational force (g), otherwise it would mean grams. Check the entire manuscript for the presence of space between the value and units (3000 g, not 3000g).

Thank you for your comment. It was corrected as suggested by the reviewer.

Line 161: How was red blood cell hemolysis calculated? ha tis the relationship between absorbance and hemolysis?

Thank you for your comment. This information was included in the manuscript.

Line 167: After going through all the manuscript there is no statistical analysis of data. Please change your results section in order to include these analyses. For instance, in the hemolysis assay the authors should analyse the data for significance.

Thank you for your comment. Statistical analysis was included as suggested by the reviewer.

4) Results

Results are superficial and do not provide a good description of important details. Some conclusions should be withdrawn from the data in the Results section so that results can be integrated.

Authors should include in this section:

Line 181: “inhibition zones can be observed around the disks” How much inhibition? Were the outcomes not quantified?

Thank you for your comment. The size of inhibition zones was included in the manuscript.

Line 189: What about C. albicans FICI? The authors must describe all data from Table 1 (FICI of S. cerevisiae and C. albicans, IC50, molecular weight and cLogP) otherwise they would not show them.

 Thank you for your comment. Data from checkerboard assay was included in another table, and we did the description of these results in section 3.3, as suggested by the reviewer. Molecular weight and cLogP description were included in section 2.2, and IC50 was described in section 3.5, as suggested by the reviewer.

Line 196: Please change "to confirm" to "to investigate" or similar.

Thank you for your comment. The change has been made as suggested by the reviewer.

Line 197: What is the role of dextrose? I can see that it has an effect in cells but the mechanism is not described. The authors should explain why they used this compound.

Thank you for your comment. This information was included in the manuscript.

Fig. 3: These results are not convincing. The quality of the images is too poor to identify cells. It is not possible to conclude that the fluorescence is emitted from within cells because they are not discernible. In addition, the magnification is not provided. When the authors obtain images of enough quality to clearly identify cells, they should count fluorescent cells and calculate their proportion in the treatments in order to present quantitative data to take conclusions.

Thank you for your comment and criticism. Indeed, there was a decrease in the quality of images when we made the panel with all of the micrographies. Therefore, we decided to send as supplementary file each image alone, to better visualization. Also, magnification was included in figure caption. About fluorescence counting, we added to the manuscript a figure with the % of rhodamine 6G efflux, obtained by measuring the fluorescence in supernatant (as described in section 2.6).

 Fig. 4: The legend should have the meaning of all abbreviations used: DGB1, DGB2,(the same applies to Fig. 5). In addition, DGB1 and DGB7 are represented with the same symbol.

Thank you for your comment. The meaning of abbreviations was included in the legend. Also, the symbols were changed.

Line 213-216: Poorly described results. The authors should withdraw conclusions from these data.

Thank you for your comment. The description of results was improved as suggested by the reviewer.

Lines 225-226: The authors should take this conclusion only after statistical analysis in order to confirm whether or not the differences are significant.

Thank you for your comment. Statistical analysis was performed as suggested by the reviewer.

 Fig. 5: The legend should include reference to replicas (n=?), statistical analysis and significance.

In all experiments reference of number of replicas should be made (all figures an tables).

Thank you for your comment. These information were included in the manuscript.

5) Discussion

This section should be totally revised in accordance with changes in the Results section.

Please revise the relevance of the findings in the light of other comparable studies.

Authors should include in this section:

Line 275: “DGB3 and DGB7 are the most hydrophobic substances” References are needed to support your statement.

Thank you for your comment. When correcting the manuscript, we observed an error when we calculate the FICI of DGB3. We adjusted it, and then we had to change this part of discussion. Therefore, this sentence was excluded from the manuscript.

Line 297: “a transporter homologue to those found in pathogenic fungi” References are needed to support your statement

Thank you for your comment. References were added to this line.

Line 299: There is no description of results with C. albicans in the results, please review this.

Thank you for your comment. This information was included and pointed in the manuscript.

6) Conclusion

This section should be totally revised in accordance with changes in the Results section.

Line 317: Please check all manuscript for in vivo in italic

Thank you for your comment. “In vivo” was corrected to “in vivo” as suggested by the reviewer.

Reviewer 2 Report

The authors described the functions of digoxin derivatives on the Saccharomyces cerevisiae. There were some problems.

1.     The title of S. cerevisiae should be Saccharomyces cerevisiae. not abbreviation.

2.     Why chose the digoxin derivatives?

3.     Figure 1. Structure of digoxin and derivatives, There were a circle in structure of digoxin

4.      Many figures were missing.

5. Reference 14 has no journal name.

Author Response

Answers and comments for Reviewer 2:

The authors described the functions of digoxin derivatives on the Saccharomyces cerevisiae. There were some problems.

  1. The title of  cerevisiaeshould be Saccharomyces cerevisiae. not abbreviation.

Thank you for your comment. The title was corrected as suggested by the reviewer.

  1. Why chose the digoxin derivatives?

Thank you for your question. Digoxin derivatives were chosen because they inhibit Na, K- ATPase. Since several MDR transporters of pathogenic fungi are also ATPases, we wondered if the derivatives could inhibit these transporters.

  1. 3.Figure 1. Structure of digoxin and derivatives, There were a circle in structure of digoxin

Thank you for your comment. We put the circle to highlight “R”, the part of digoxin that was substituted in derivatives structures.

  1.   Many figures were missing.

Thank you for your comment. We made a mistake when the manuscript was uploaded. Figures are already available for evaluation.

  1. Reference 14 has no journal name.

Thank you for your comment. This information was included as pointed out by the reviewer.

Reviewer 3 Report

In lines 35 and 36 it is important to mention that fluconazole was one of the first azole drugs of choice approved by the FDA for fungal infections, with the exception of fungi that cause onychomycosis.

Lines 47-50. It is important to explain that Saccharomyces cerevisiae is a ubiquitous yeast, but nevertheless, the incidence of invasive infection produced by these fungi has increased significantly in recent decades.

Line 71. Insert the institution to which Dr. Theodore White belongs.

Lines 76 and 77 it is important to recognize whether fluconazole is of generic or patent origin.

Lines 88 - 89. Mention which microbial dilution method was used to evaluate the antifungal activity and if there is any international standard for it.

Lines 101 - 102. It would be interesting to place in the text of the article some of the photographs taken as supporting material and write a brief explanation at the bottom of the same.

Line 291. the word in vivo should be in italics.

It would be interesting if the authors would indicate the methods that were used to identify the yeasts used in the study.

Author Response

Answers and comments for Reviewer 3:

In lines 35 and 36 it is important to mention that fluconazole was one of the first azole drugs of choice approved by the FDA for fungal infections, with the exception of fungi that cause onychomycosis.

Thank you for your comment. This information was included as suggested by the reviewer.

Lines 47-50. It is important to explain that Saccharomyces cerevisiae is a ubiquitous yeast, but nevertheless, the incidence of invasive infection produced by these fungi has increased significantly in recent decades.

Thank you for your comment. This information was included as suggested by the reviewer.

Line 71. Insert the institution to which Dr. Theodore White belongs.

Thank you for your comment. This information was included as suggested by the reviewer.

Lines 76 and 77 it is important to recognize whether fluconazole is of generic or patent origin.

Thank you for your comment. This information was included as suggested by the reviewer.

Lines 88 - 89. Mention which microbial dilution method was used to evaluate the antifungal activity and if there is any international standard for it.

Thank you for your comment. More information regarding the method was included as suggested by the reviewer.

Lines 101 - 102. It would be interesting to place in the text of the article some of the photographs taken as supporting material and write a brief explanation at the bottom of the same.

Thank you for your comment. We made a mistake when the manuscript was uploaded. Figures are already available for evaluation and photographs are present in the final version.

Line 291. the word in vivo should be in italics.

Thank you for your comment. “In vivo” was corrected to “in vivo” as suggested by the reviewer.

It would be interesting if the authors would indicate the methods that were used to identify the yeasts used in the study.

Thank you for your comment. This information was included in the manuscript as suggested by the reviewer.

Round 2

Reviewer 1 Report

In this version of the manuscript, the authors successfully addressed most of the issues, however, some issues were not answered or were not addressed convincingly (see below). English language still needs revision. This version of the manuscript still requires revision to be accepted for publication.

Abstract
All issues replied. Some new issues:
Please remove the abbreviations or define them in the abstract (DGB, FICI). The abstract should stand alone without any reference to the manuscript.

Line 26
It is wiser to avoid such definite conclusions. It is more advisable to say “Results suggest…”.

Introduction
All issues replied.

Materials and methods
All issues replied. Some unrepplied issues:
Lines 84-85
The genotype of AD/124567 is incompete. The mating type and the genetic determinants for pdr5p overexpression are missing. Check also for the genotype of AD/1234567.

Line 188
The authors did not explicitly mention a centroifugation step here. Did they centrifuge the samples? If yes, what were the conditions?

Line 197
The units of zymolyase used are still missing. This information is important in order to characterise the enzymatic reaction and to ensure cell wall hydrolysis.

Results
All issues replied. Some unrepplied issues:
Supplementary photos.
AD/124567 without treatment and without glucose yield different fluorescence in cells. See photos S2 and S4. How do the authors explain this?
S5 and S6 were treated with glucose? It is not clear.

Fig. 3
These data are still not convincing. The photos are the same as in the previous version. However, the photos supplied as supplementary material have enough quality. The authors should provide a way to treat these photos in order to have quality for publishing otherwise they should remove the figure. In addition, as suggested before, the authors should provide quantitative data by presenting the percentage of cells with fluorescence. Visual inspection is not enough when there is the possibility of counting ~100 cells in each treatment and calculate the % of cells wit fluorescence.
In addition, Fig. 4 provides strong evidence of the efflux of R6G only when these results are compared with the fluorescence of cells. I agree that results in Fig. 4 are evidence suggesting inhibition of efflux by DGB derivatives however, these results should be complemented with the quantitation of fluorescent cells.

Fig. 4
The name of the strain is missing.

Section 3.5
The presentation of results is still poor. The authors should take conclusions from the results. This should be applied to all sections of results.

Discussion
The discussion section was not reformulated. In the Results section, authors should present all results, make comparisons and take conclusions, In the Discussion section, authors should compare their results with the literature, take more conclusions in the context of the literature and highlight the most relevant findings. Nevertheless, this version might be suitable for publishing.

Author Response

In this version of the manuscript, the authors successfully addressed most of the issues, however, some issues were not answered or were not addressed convincingly (see below). English language still needs revision. This version of the manuscript still requires revision to be accepted for publication.

Abstract
All issues replied. Some new issues:
Please remove the abbreviations or define them in the abstract (DGB, FICI). The abstract should stand alone without any reference to the manuscript.

Thanks for your suggestions and now, we described what means the abbreviations in the abstract.

Line 26
It is wiser to avoid such definite conclusions. It is more advisable to say “Results suggest…”.

We accepted your suggestion and changed the word “show”.

Introduction
All issues replied.

Materials and methods
All issues replied. Some unrepplied issues:
Lines 84-85
The genotype of AD/124567 is incompete. The mating type and the genetic determinants for pdr5p overexpression are missing. Check also for the genotype of AD/1234567.

The missing information was included as suggested by the reviewer.

Line 188
The authors did not explicitly mention a centroifugation step here. Did they centrifuge the samples? If yes, what were the conditions?

The conditions of centrifugation were described at Mat&Met (item 2.6): “…pelleted by centrifugation at 9000 rpm for 2 minutes”.

Line 197
The units of zymolyase used are still missing. This information is important in order to characterise the enzymatic reaction and to ensure cell wall hydrolysis.

The units of zymolyase were included in the manuscript as suggested by the reviewer.

Results
All issues replied. Some unrepplied issues:
Supplementary photos.
AD/124567 without treatment and without glucose yield different fluorescence in cells. See photos S2 and S4. How do the authors explain this?
S5 and S6 were treated with glucose? It is not clear.

Thanks for your inquiries and we will try to explain better to answer your questions. In fact, the information that were incubated in the presence of glucose at legend of S5 and S6 are missing (we added this information at legend of figure and in the supplementary data). We did a control (S2 and S4) using the strain that overexpresses Pdr5p, but in the absence of glucose to prove that without energy (obtained from glucose metabolism) the ABC transporter (Pdr5p) cannot work therefore, the R6G remains inside the cells, that is very similar when we compare with control using the strain AD1234567, that doesn’t have an overexpression of Pdr5p at plasma membrane, so, the fluorophore remains at cytoplasmic side.

Fig. 3
These data are still not convincing. The photos are the same as in the previous version. However, the photos supplied as supplementary material have enough quality. The authors should provide a way to treat these photos in order to have quality for publishing otherwise they should remove the figure. In addition, as suggested before, the authors should provide quantitative data by presenting the percentage of cells with fluorescence. Visual inspection is not enough when there is the possibility of counting ~100 cells in each treatment and calculate the % of cells wit fluorescence.
In addition, Fig. 4 provides strong evidence of the efflux of R6G only when these results are compared with the fluorescence of cells. I agree that results in Fig. 4 are evidence suggesting inhibition of efflux by DGB derivatives however, these results should be complemented with the quantitation of fluorescent cells.

We understand your points and concerns about the pictures but, we will try to explain all idea in the construction of these pictures (in the same figure) using the data of Figure 4 and table 2, that must be analyzed together. The pellets used to check if the fluorescence is present or not of compounds are only for a qualitative observation and of course, to guide the readers to see the phenomenon of ABC transporter action in loco, using the cells directly. Since we need to centrifuge the samples, to be sure that we are not going to have cells at supernatant, the pellet is compact, and it is not uniform that allow us to count free cells with no trouble. Therefore, to quantify the fluorescence intensity precisely, present at supernatant and result of the action of the active transporter on plasma membrane, we used the data of fluorescence measurement where we can quantify numerically all conditions, in the presence or absence of digoxin derivatives. So, in our opinion, data obtained after cell counting using the pellets would not bring us accurate information as those shown in figure 4 and table 2.

Fig. 4
The name of the strain is missing.

Thanks for your information. The strain used was placed in the legend of figure now.

Section 3.5
The presentation of results is still poor. The authors should take conclusions from the results. This should be applied to all sections of results.

Results description was improved as suggested by the reviewer.

Discussion
The discussion section was not reformulated. In the Results section, authors should present all results, make comparisons and take conclusions, In the Discussion section, authors should compare their results with the literature, take more conclusions in the context of the literature and highlight the most relevant findings. Nevertheless, this version might be suitable for publishing.

The discussion section was improved as suggested by the reviewer.

Reviewer 2 Report

The paper can be accepted after revision.

Author Response

Thanks for your review and acceptance of our manuscript. The mansucript was reviewed again in order to improve the english language.